# Decellularized Bovine Skeletal Muscle Scaffolds: Structural Characterization and Preliminary Cytocompatibility Evaluation

**DOI:** 10.3390/cells13080688

**Published:** 2024-04-16

**Authors:** Luana Félix de Melo, Gustavo Henrique Doná Rodrigues Almeida, Felipe Rici Azarias, Ana Claudia Oliveira Carreira, Claudete Astolfi-Ferreira, Antônio José Piantino Ferreira, Eliana de Souza Bastos Mazuqueli Pereira, Karina Torres Pomini, Marcela Vialogo Marques de Castro, Laira Mireli Dias Silva, Durvanei Augusto Maria, Rose Eli Grassi Rici

**Affiliations:** 1Graduate Program in Anatomy of Domestic and Wild Animals, University of São Paulo, São Paulo 03828-000, Brazil; biologafelix@gmail.com (L.F.d.M.); ana.carreira@ufabc.edu.br (A.C.O.C.); roseeli@usp.br (R.E.G.R.); 2Graduate Program of Medical Sciences, College of Medicine, University of São Paulo, São Paulo 03828-000, Brazil; felipe.azarias@usp.br; 3Center of Human and Natural Sciences, Federal University of ABC, Santo André 09210-170, Brazil; 4Department of Pathology, School of Veterinary Medicine and Animal Science, University of São Paulo, São Paulo 03828-000, Brazil; csastolfi@gmail.com (C.A.-F.); ajpferr@usp.br (A.J.P.F.); 5Graduate Program in Structural and Functional Interactions in Rehabilitation, Postgraduate Department, University of Marília (UNIMAR), Marília 17525-902, Brazil; elianabastos@unimar.br (E.d.S.B.M.P.); karinatorrespomini@gmail.com (K.T.P.); marcelavialogo@hotmail.com (M.V.M.d.C.); lairadias@outlook.com.br (L.M.D.S.); 6Development and Innovation Laboratory, Butantan Institute, São Paulo 05585-000, Brazil; durvanei@usp.br

**Keywords:** skeletal muscle, tissue engineering, muscle regeneration, scaffolds, biomaterials

## Abstract

Skeletal muscle degeneration is responsible for major mobility complications, and this muscle type has little regenerative capacity. Several biomaterials have been proposed to induce muscle regeneration and function restoration. Decellularized scaffolds present biological properties that allow efficient cell culture, providing a suitable microenvironment for artificial construct development and being an alternative for in vitro muscle culture. For translational purposes, biomaterials derived from large animals are an interesting and unexplored source for muscle scaffold production. Therefore, this study aimed to produce and characterize bovine muscle scaffolds to be applied to muscle cell 3D cultures. Bovine muscle fragments were immersed in decellularizing solutions for 7 days. Decellularization efficiency, structure, composition, and three-dimensionality were evaluated. Bovine fetal myoblasts were cultured on the scaffolds for 10 days to attest cytocompatibility. Decellularization was confirmed by DAPI staining and DNA quantification. Histological and immunohistochemical analysis attested to the preservation of main ECM components. SEM analysis demonstrated that the 3D structure was maintained. In addition, after 10 days, fetal myoblasts were able to adhere and proliferate on the scaffolds, attesting to their cytocompatibility. These data, even preliminary, infer that generated bovine muscular scaffolds were well structured, with preserved composition and allowed cell culture. This study demonstrated that biomaterials derived from bovine muscle could be used in tissue engineering.

## 1. Introduction

Skeletal striated musculature provides mobility, allows locomotion and individual survival in the environment, and provides strength and resistance to the organism [1]. Severe muscle injuries that lead to excessive muscle mass loss, which can occur by trauma, infarction, or tumor resection, can cause muscle degeneration and scar tissue formation, compromising organ functionality [2]. These injuries are not restricted to humans but are also present in other species, affecting companion animals like dogs and cats, farm animals, and even valuable racing animals like prize horses [3]. Due to great anatomic diversity among the species, there are several therapeutic approaches that have mixed results and are ineffective in more severe cases [4]. In humans, treatment options based on rehabilitation, repair of the injured area, and even replacement of affected muscle segments are among the conventional approaches; however, in severe cases, they are unsuccessful [5].

Autologous muscle graft (AMG) is the most recommended and used approach in orthopedics to reestablish muscle integrity [6]. However, AMG is related to several complications, such as the risk of graft failure, prolonged surgical exposure time, development of opportunistic infections, and tissue necrosis [7,8]. The use of allografts or xenografts is even more restricted as the greater risk of rejection and immunosuppressive therapy are related to several complications, such as kidney failure, hypertension, and gastrointestinal and liver problems, among others [9].

In order to overcome such limitations, studies have focused on in situ stem cells and satellite cell transplantation for muscle injuries; however, clinical studies are limited, and their viability is restricted [10,11]. Therefore, considering tissue engineering principles, studies have begun focusing on biomaterial application, which, combined with cells and bioactive molecules, can promote more effective results [12,13,14].

In this context, 3D scaffolds provide functional and structural support for cell anchorage, proliferation, and differentiation, enhancing the tissue microenvironment complexity, which leads to a more effective reestablishment of the injured area [15]. Several synthetic and natural biomaterials have been proposed to stimulate muscle cell proliferation or promote muscle regeneration, but up until now, no material has been able to mimic the native extracellular matrix complexity, which has biomechanical and biochemical properties that provide intense tissue-specific molecular signaling [16]. Hence, in order to obtain extracellular matrices (ECM) with high biological quality, decellularization has been proposed to generate 3D acellular structures with preserved ECM proteins and with low or no immunogenicity [17]. The skeletal striated musculature matrix has specific structural organization and biomechanical properties that contribute to muscle formation, which are an excellent candidate for in vitro recellularization studies and in vivo regenerative potential [18].

Previous studies have applied non-tissue-specific ECM scaffolds for muscle repair, such as commercial decellularized porcine small intestinal submucosa (SIS), urinary bladder, and even dermis. Although some results to fill volumetric muscle loss were significant, such biomaterials lack muscle-specific components [18]. Differently from other ECM-based scaffolds, acellular muscle matrices can provide signaling instructions for cell repopulation through their specific organization and composition, which differs from every other tissue [19]. In addition, studies have already demonstrated the decellularized skeletal muscle matrix immunomodulatory properties, which provide a more suitable microenvironment for muscle cells to proliferate and establish [20,21].

The majority of available decellularized muscle scaffolds are from small animals such as mice, rats, or even rabbits, which are suitable for experimental purposes, but in terms of applicability and clinical application, they have limited potential [18,22]. Regarding translational application, large livestock animals are a great source of tissue for scaffold production; due to food production, there is a great availability, and their size is more compatible for use in human and veterinary clinics [23,24]. Acellular scaffolds derived from porcine skeletal muscle have shown promising results in vitro, being able to support cell culture. This corroborates the initiative for the development of biomaterials derived from other large animals, such as the bovine, which has been widely used for other biotechnological purposes, increasing the availability of ECM-based biomaterials for muscle regenerative approaches [25,26,27].

Therefore, this study aimed to produce and characterize biological scaffolds derived from bovine skeletal musculature, evaluating the decellularization efficiency regarding cell removal and ECM component preservation. In addition, preliminary in vitro cytocompatibility assays using bovine fetal myoblasts were performed to infer the scaffold application in 3D cell culture.

## 2. Materials and Methods

### 2.1. Sample Acquisition

Femoral bicep muscle samples (n = 5 per animal age) of 2 × 2 × 2 cm in size from fetuses of 4 and 5 months of age and adults of Nellore cattle (*Bos tauros indicus*) in age for slaughter were obtained from Barra Mansa slaughterhouse, Sertaozinho, São Paulo State, Brazil. They were transported to the Faculty of Veterinary Medicine and Animal Science—FMVZ, Anatomy of Domestic and Wild Animals sector of the University of São Paulo—USP. The study was submitted and approved by the Animal Use Ethics Committee (protocol n° 5856110329). The methodology is summarized in Figure 1.

### 2.2. Decellularization

Bovine muscle samples were submitted to decellularization, which consisted of immersion in solutions and intense agitation for seven days. At first, samples were washed in distilled water (dH_2_O) three times for 5 min and then washed with sodium chloride (NaCl) 1 M overnight. The samples were immersed in EDTA for 2 h and washed in dH_2_O three times for 5 min. Afterward, the samples were immersed and agitated in 2% SDS for 72 h with changes each 12 h. Then, the samples were immersed in 1% Triton X-100 for 6 h and then washed in dH_2_O for 48 h with changes each 12 h. The decellularized scaffolds were stored in PBS 1X for further analyses.

### 2.3. 4,6-Diamidino-2-Fenilindole (DAPI) Staining

In order to assess the nuclei presence in decellularized muscle scaffolds, DAPI staining was performed. Native and decellularized samples were frozen in Tissue Plus O.C.T. (Fisher Health Care, Houston, TX, USA) and microsectioned using a cryostat (CM1860 model, LEICA Biosystems, Baden-Wurttemberg, Germany). The slices were stained with DAPI solution (1:10,000) for 15 min. Then, they were washed with PBS 1X for analysis using fluorescent microscopy (Nikon ECLIPSE 80I, CADI FMVZ-USP, Tokyo, Japan).

### 2.4. Total Genomic DNA Quantification

After decellularization, the samples were submitted to total genomic DNA quantification to confirm the process efficiency. Genomic DNA was isolated from 30 mg of decellularized muscle using the Illustra Tissue and Cells Genomic Prep Mini Spin Kit (GE Healthcare), according to the manufacturer’s recommendations. Samples were digested with Proteinase K and lysis buffer from the kit at 56 °C for 2 h. The purified gDNA was analyzed using a spectrophotometer at 260 nm (Nanodrop, Thermo Scientific, Waltham, MA, USA). For all quantifications, three replicates were analyzed.

### 2.5. Histological Analysis

Native and decellularized samples were fixed in 4% paraformaldehyde (PFA) for 48 h, then dehydrated in crescent ethanol concentration, from 70% to 100%, diaphanized in xylol, and embedded in paraffin. Microsections of 5 µm (n° RM2265; LEICA) were stained with Hematoxylin & Eosin (HE) to evaluate the sample structure general morphology; Masson’s trichrome to evaluate the collagen presence and organization; Picrosirius Red to evaluate the preservation of different types of collagen fibers, according to their thickness and location; Alcian blue to evaluate the presence of glycosaminoglycans and Colloidal Iron to evaluate the presence of acid mucopolysaccharides. Slides were photographed and analyzed using a light microscope (Nikon ECLIPSE 80I, CADI FMVZ-USP).

### 2.6. Ultrastructural Analysis by Scanning Electronic Microscopy (SEM)

SEM analysis was performed to evaluate the native and decellularized muscle sample three-dimensional (3D) structure. The tissues were fixed in 10% formaldehyde for 48 h, washed with distilled water (5x for 5 min), stored in 70% alcohol overnight, and dehydrated in ethanol (80%, 90%, and 100% 2x, 10 min each). The samples were dried in a critical point device with CO_2_ LEICA EM CPD 300, glued with carbon glue to aluminum metal bases (stubs) and metalized (“sputting”) with gold in the EMITECH K550 metalizing device, and then analyzed and photodocumented in a scanning electron microscope (LEO 435VP) at the Advanced Center for Diagnostic Imaging—CADI-FMVZ-USP.

### 2.7. Immunofluorescence Analysis

Native and decellularized samples were embedded in O.C.T. (Optimal Cutting Temperature) and frozen at −150 °C. Sections of 12 μm were fixed in 4% PFA for 10 min, dried at room temperature for 5 min, incubated in PBS + 10% goat serum for 1 h, and incubated with primary antibodies overnight at 4 °C in a dark, humid chamber. The antibodies and dilutions used are listed in Table 1. Subsequently, the slides were washed (3x, 5 min each) with PBS + 1% goat serum at room temperature, incubated with the secondary antibody for 1 h, and protected from light. The slides were washed in PBS solution (3x, 5 min), incubated with DAPI (1:10,000) for 10 min at room temperature, and washed again with PBS (3x, 5 min). They were analyzed under a confocal microscope (FV1000 Olympus IX81, Olympus, Tokyo, Japan) with 20x and 40x objectives on the Nikon Eclipse E-80i fluorescence light microscope from the Advanced Center for Image Diagnosis—CADI-FMVZ-USP.

### 2.8. Water Absorption Analysis

Cubic-shaped samples of 2 cm^3^ from decellularized muscle scaffolds were used for this assay, which was carried out according to the Water Absorption ASTM D570 Guideline. The samples were submitted to a 24-h immersion. For that, samples were immersed in a petri dish with distilled water and stored in a humidified oven at a temperature of 37 °C at 5% CO_2_. The measurement was performed for 24 h. Then, the samples were removed from the water, placed on a glass slide to remove excess water, and immediately weighed on an analytical balance.

### 2.9. Cell Isolation and Characterization

Fetal muscle tissue was washed in phosphate buffer saline (PBS 1X) + 4% streptomycin and penicillin in a 15 mL falcon tube under manual agitation for approximately 10 min to ensure the material was clean. Then, the tissues were fragmented into explants to obtain muscle-derived cells. The cells were cultured in Alpha-MEM/F12 media supplemented with 10% BFS and 1% antibiotics at 37 °C and 5% CO_2_. P1 to P3 cells were used for the experiments.

In order to characterize the myoblasts, immunofluorescence for MyoD, vimentin, and N-cadherin were performed. Cells were fixed in 4% PFA for 10 min, dried at room temperature for 5 min, incubated in PBS + 10% goat serum for 1 h, and incubated with primary antibodies overnight at 4 °C in a dark, humid chamber. The antibodies and dilutions used are listed in Table 1. Then, the slides were washed (3x, 5 min each) with PBS + 1% goat serum at room temperature, incubated with the secondary antibody for 1 h, and protected from light. The slides were washed in PBS solution (3x, 5 min), incubated with DAPI (1:10,000) for 10 min at room temperature, and washed again with PBS (3x, 5 min). They were analyzed under a confocal microscope (FV1000 Olympus IX81, Japan) with 20x and 40x objectives on the Nikon Eclipse E-80i fluorescence light microscope from the Advanced Center for Image Diagnosis—CADI-FMVZ-USP.

### 2.10. Scaffold Sterilization

Scaffold fragments were dried in a critical point device with CO_2_, passed through a series of 70% alcohol three times for 3 min, and placed in ultraviolet (UV) light for 10 min before starting the recellularization protocol.

### 2.11. Three-Dimensional Culture of Fetal Muscle-Derived Cells

After the characterization, the scaffolds were immersed in Alpha-MEM supplemented with 10% BFS and incubated at 37 °C and 5% CO_2_ for 48 h to attest to the biomaterial sterility. Afterward, 5 × 10^4^ bovine fetal skeletal muscle-derived cells were seeded under the same culture condition for 4, 8, and 10 days. Then, the fragments with seeded cells were fixed for SEM analysis.

### 2.12. Statistical Analysis

Morphometry was performed using point tests and presented a normal distribution according to the Shapiro–Wilk test. The data obtained were analyzed using Student’s *t*-test with Tukey’s post-test, using the statistical program GraphpadPrism (Version 7.0). A significance level of 5% was adopted (*p* < 0.05).

## 3. Results

### 3.1. Decellularization Process Evaluation

Bovine musculature is very heterogeneous, and muscle and fat cell proportion, as well as the biomechanical properties, influence muscle tissue structure [28]. For that reason, the samples were obtained and standardized from the femoral bicep muscle due to its representative size and weight in the animal, the shear force being between the threshold of slightly soft and slightly hard, its low to medium financial cost, and significant water retention capacity [29]. Macroscopically, after the decellularization, muscle samples were whitish and translucent compared to native samples due to cell and blood removal (Figure 2A). Microscopically, DAPI staining revealed the absence of nuclei compared to the native sample, attesting to the decellularization efficiency (Figure 2B–E). In addition, total genomic DNA quantification demonstrated a reduction of 95.24% in the DNA content in the decellularized samples (Figure 2F).

In order to attest to the fluid uptake ability of the samples, a water absorption assay was carried out (Figure 3). Dried scaffolds were stored in sealed plates, protected from light and humidity in the fridge at 4 °C. The assays could be conducted monthly for six months after drying. After a 24-h immersion in distilled water, the wet scaffolds were weighed and compared to the dry samples’ previous weight. The analysis demonstrated a significant increase of almost 400% in the sample weight, attesting to the scaffolds’ water absorption capacity (Figure 3).

### 3.2. Structural and Ultrastructural Characterization

ECM component preservation is a major characteristic of functional biological scaffolds, which includes structural and glycoadhesive proteins that allow cell adhesion, proliferation, and survival [24]. Therefore, histological, immunofluorescence, and ultrastructural analyses were applied to perform a full characterization of generated bovine muscle scaffolds in terms of structure, composition, and three-dimensionality (Figure 4, Figure 5 and Figure 6). Concerning general ECM structure and components, HE staining revealed multinucleated cylindrical cells and elliptical nuclei on the periphery of the fibers close to the plasma membrane in the native muscle tissue, which characterize skeletal muscle morphology (Figure 4A). In addition, the perimysium and endomysium interspersed with vessels were notable. In the decellularized samples, the absence of cells and connective tissue preservation were observed, and the tissue remained structured according to muscle fiber organization (Figure 4B).

Colloidal Iron and Alcian Blue staining were used to assess the preservation of proteoglycans and glycosaminoglycans, respectively (Figure 4C–F). In native samples, muscle fibers are highlighted in yellow, while the proteoglycan content is discretely stained in pinkish red (Figure 4C), while in decellularized samples, a more pronounced pinkish red content can be observed, demonstrating the preservation of such components after cellular removal (Figure 4D). The same pattern is observed in Alcian Blue staining; light blue tones are observed among muscle fibers, which are highlighted by their peripheral nuclei (Figure 4E). In the decellularized samples, light blue tones are still evident, even with the absence of cells, which infers the GAG content preservation (Figure 4F).

Lastly, total collagen content was assessed using Masson’s trichrome staining (Figure 4G,H). In native samples, muscle fibers are stained in red tones and black nuclei, while the collagen from the connective tissue is stained in purple (Figure 4G). After the decellularization, cellular components are absent, and the collagen scaffold is still structured (Figure 4H).

After the general structure evaluation, a deeper investigation of collagen composition, distribution, and 3D structure was performed, as collagen is the most abundant component of perimysium and endomysium connective tissue, and it is responsible for tissue organization and stability [30]. For that, Picrosirius Red staining was applied to evaluate collagen fiber type distribution in native and decellularized bovine muscle samples (Figure 5A–H). Under polarized light microscopy, the distinction of different collagen types by fiber diameter is possible since fibers with reddish and yellowish tones are related to thick collagen fibers and greenish ones are related to thin fibers [31,32,33]. Both types of fibers were preserved after the decellularization process and the morphoquantitative analysis demonstrated no significant difference between both groups (Figure 5I).

In order to assess the scaffolds’ three-dimensionality, SEM analysis was carried out (Figure 5J–Q). In native samples, myotube-forming muscle bundles can be observed surrounded by endomysium and perimysium, which are characteristic of skeletal muscle compartmentalization (Figure 5J–M). After decellularization, the native muscle structure becomes loosened due to the absence of myotubes, but it is noticeable that there are no signs of degradation or destruction (Figure 5N–Q). The presence of thick and thin fibers and the maintenance of empty circular spaces, which were previously occupied by myotubes, highlight that muscle scaffolds remained structured. Comparing Picrosirius histological findings with SEM images, it is clear that in decellularized samples, the fibers are not organized as in the native tissue. This finding is correlated to the absence of cells, which are no longer anchored and do not play stress forces on ECM fibers anymore, favoring events such as collagen self-assembly and thicker bundle formation, as shown in Figure 5P.

ECM has a great number of bioactive components that perform different roles in the tissue microenvironment. In order to evaluate ECM molecule preservation in the decellularized scaffolds, one protein of each macrogroup was chosen for immunofluorescence analysis (Figure 6). The chosen proteins were fibronectin (adhesive glycoprotein), elastin (elastic fiber), perlecan (proteoglycan), and hyaluronic acid (glycosaminoglycan). In native samples, there is a more intense presence of fibronectin close to blood vessels and interspersed in bundles (Figure 6A). In decellularized tissue, fibronectin is present and more loosened due to cell absence, but with no signs of degradation (Figure 6D).

Regarding the elastic fibers, which are a major ECM component related to tissue stiffness and flexibility, elastin immunolocalization was assessed. Both in native and decellularized samples, elastin presence was more intense in the perimysium region and in blood vessels (Figure 6G–L). In addition, perlecan and hyaluronic acid immunolocalization was evaluated (Figure 6M–X). Both proteins remained preserved in decellularization samples, inferring proteoglycan and glycosaminoglycan preservation.

### 3.3. Cellular Assays

As a preliminary result to attest to the bovine muscle scaffold cytocompatibility, fetal bovine myoblasts were cultured on the scaffolds for 4, 8, and 10 days in order to assess their ability to anchor and proliferate on the scaffolds (Figure 7 and Figure 8). In order to obtain myoblasts in second myogenesis, samples were obtained and standardized from bovine fetuses of five months of age according to crown–rump measurements [34]. In order to characterize the myoblast population, three markers were chosen: MyoD, N-cadherin, and Vimentin. MyoD is a classic myoblast marker and plays an important role in skeletal muscle fiber differentiation during development [35]. N-cadherin, an adhesion molecule, is also an important marker and is highly expressed by skeletal muscle precursor cells [36]. Moreover, vimentin, an intermediary filament component of the cytoskeleton, is expressed predominantly in myoblasts, which is replaced by desmin in adult myotubes [37]. The cells presented strong markers for all three markers, providing enough data to attest to the myoblast population profile (Figure 7).

After the characterization, fetal bovine fibroblasts were cultured on the muscle scaffolds and analyzed for 4, 8, and 10 days to assess cell adhesion and establishment (Figure 8). Following the evaluated days, it was possible to observe cell adhesion and anchoring on the scaffolds’ surface, exhibiting pronounced elongations of the cell membranes, characteristic of cell adhesion and migration. Comparing day 4 with the other analyzed periods, there is a notable increase in cell density on the scaffolds, mainly comparing Figure 4A,B, in which the cells are more interspaced with Figure 4E,F, in which a more dense layer can be observed, suggesting cell proliferation and survival. Although these are preliminary data, the assay attested to the scaffolds’ ability to allow myoblast culture, providing structural and functional conditions for cell adhesion, migration, and proliferation. More experiments are required to characterize cellular behavior on bovine muscle scaffolds.

## 4. Discussion

Application of decellularized scaffolds in 3D muscle cell culture and muscle repair studies has demonstrated satisfactory results that highlight the high biological potential of these biomaterials [18,19,25,27]. Although several studies have already described and standardized the decellularization for muscle tissue, most of them used samples from small lab animals such as rats, mice, and rabbits, which, considering translational science, have limited application [18]. Ideally, scaffolds derived from human muscle tissue would be the best option to use in clinical practice due to allogenic compatibility; however, this scenario is not realistic, as human sample acquisition is restricted by several bioethical drawbacks, becoming an enviable source for large-scale purposes [38]. In addition, studies have already attested that xenogeneic acellular matrices, when implanted in vivo when well produced, generate little or no immunological response in the host, overcoming immune limitations for their application [39,40].

As mentioned, few studies have used muscle tissue from large animals to produce decellularized scaffolds, being restricted to porcine tissue [18]. A recent study evaluated different decellularization protocols for porcine skeletal muscle, analyzing DNA content and ECM preservation [41]. Among the evaluated protocols, the SDS-based one was the most promising, showing the best results concerning ECM composition and decellularization efficiency, similar results to those found with our protocol. Other studies also showed that porcine skeletal muscle-derived scaffolds and hydrogels were cytocompatible and allowed human C2C12 myotube formation and survival, which demonstrated that large animal muscle scaffolds can support human cell culture [41,42]. Despite significant results, porcine-derived biomaterials have an imperative bioethical disadvantage due to religious beliefs. Porcine-based biomedical therapies and devices suffer resistance from a great number of patients, thus limiting their application since a large percentage of the population does not consume porcine products [43,44].

Based on that fact, new sources for biomedical implants and biomaterials are required to expand the population’s access to this sort of treatment and biotechnological therapies. Bovine tissue-derived biomaterials such as the pericardium patch and bovine heart valves have been considered reliable and high-standard tools in surgical practice, being widely used in cardiovascular surgical procedures [45,46]. Another factor for bovine tissue prospection as a biological source for biomaterial production is their great availability; the slaughter of cattle is a recurrent practice, and, with such biotechnological potential, beef could be converted into scaffolds for the development of biological dressings or substrate for bioactive hydrogels production [47,48]. Only one study, up until now, explored the production of decellularized scaffolds from bovine skeletal muscle, which were associated with polycaprolactone (PCL), to develop a hybrid biocomposite that allowed supported satellite cell differentiation [49]. The matrices were obtained from bovine tail muscles, which are a limited source of tissue for biotechnological purposes. These results encouraged the development of biomaterials derived from bovine muscle ECM, which may present great biological activity for muscle stimulation [49].

Differently from other tissues, skeletal muscle tissue has a great variety considering the region of origin, which impacts ECM composition, tensile strength, and availability [50,51]. Considering this, our study aimed to establish an efficient and low-cost protocol to produce bovine muscle-derived scaffolds, which can be applied to further tissue-engineering approaches. The proposed protocol was effective in removing the cellular content as long as it preserved muscle tissue structure and its components. Skeletal striated muscle matrix microarchitecture is complex and compartmentalized, which contributes not only to muscle resistance and contractility but also to tissue repair in situations of injury [52].

ECM component preservation is a key factor for scaffold bioactivity; each group from the muscular matrisome plays a different role in muscular physiology [53]. Structural components such as collagen and elastic fibers were highly preserved in our biomaterials, demonstrating that major molecules related to muscle tension strength, elasticity, and rigidity were present, which maintained the muscle 3D organization [53,54]. Glycoprotein presence, such as fibronectin, was also assessed, as these proteins are related to cell adhesion and muscle stem cell regulation niche [55]. Glycoproteins such as laminin and fibronectin are key components that allow interaction between the intracellular and extracellular microenvironment through integrin binding, which allows molecular communication through mechanotransduction [56]. These glycoproteins, along with proteoglycans such as syndecan, decorin, perlecan, and other basement proteins such as type IV collagen, are related to a balance between stem cell differentiation and self-renewal, regulating the skeletal muscle regenerative capacity [57,58]. In addition, other non-fibrillary protein groups, glycosaminoglycans (GAGs), such as hyaluronic acid, are related to muscle repair initial steps, being a transient component related to ECM remodeling after a muscle injury [59]. Our results attested to the presence of these main proteins, which infer their biological activity.

It is noteworthy that cell–cell and cell–ECM interaction increases significantly in 3D microenvironments, affecting cell behavior related to migration, proliferation, stem cell self-renewal, and differentiation, which provide more reliable data compared to the conventional 2D culture system [60]. Adequate scaffold porosity is required to interconnect these pore structures and mimic tissue compartments, which contribute to cellular communication and facilitate gas and nutrient exchange, preventing hypoxia and tissue necrosis, mainly in the scaffold’s inner portions [61,62]. Results obtained from SEM analysis and the water absorption assay demonstrated that bovine muscle scaffolds have a high fluid uptake ability, with well-structured interspaced structures, which allow fluid transfer.

Moreover, such structures may be able to support cell culture to validate their cytocompatibility properties, but despite their application for in vitro models, in vivo implantation, or hydrogel production, such materials must be correctly sterile and non-cytotoxic [63,64]. Previous studies have cultured myoblasts and myotubes on scaffolds to attest to their cytocompatibility, assessing their ability to adhere, survive, and proliferate [41,42]. Our results, although preliminary, attested that bovine myoblasts, which are primary and immature cells, were able to interact with the scaffold, presenting cellular projections after 10 days of culture. These findings inferred migratory and proliferative movements along the muscle scaffold. Altogether, our data demonstrated that these scaffolds have a preserved composition, well-structured 3D organization, high fluid uptake ability, and are cytocompatible. More studies must be conducted to determine cell differentiation on the scaffolds, in vivo biocompatibility and mechanical properties. Decellularized scaffold application for muscle regeneration and biomimetic reconstruction is in its infancy, but to encourage advances in the field, novel bioactive biomaterials must be generated and characterized to be applied to clinical trials and be a viable option for severe muscle injuries, which will influence directly on the patient’s life quality [65].

## 5. Conclusions

In this study, we produced and characterized the first bovine skeletal muscle-derived scaffolds, attesting their structural and compositional preservation. The generated biomaterials presented the main ECM components without signs of degradation as important ECM proteins related to tissue organization, cell adhesion, and differentiation remained intact after the decellularization. In addition, preliminary data suggested cytocompatibility properties, allowing myoblast cell adhesion and interaction with the scaffolds after 10 days of culture. This data, although preliminary, infers that such scaffolds may be utilized as biological scaffolds for muscle cell culture and potential application in skeletal muscle production. Further studies assessing muscular differentiation, gene expression, and protein quantification are required.

## Figures and Tables

**Figure 1 cells-13-00688-f001:**
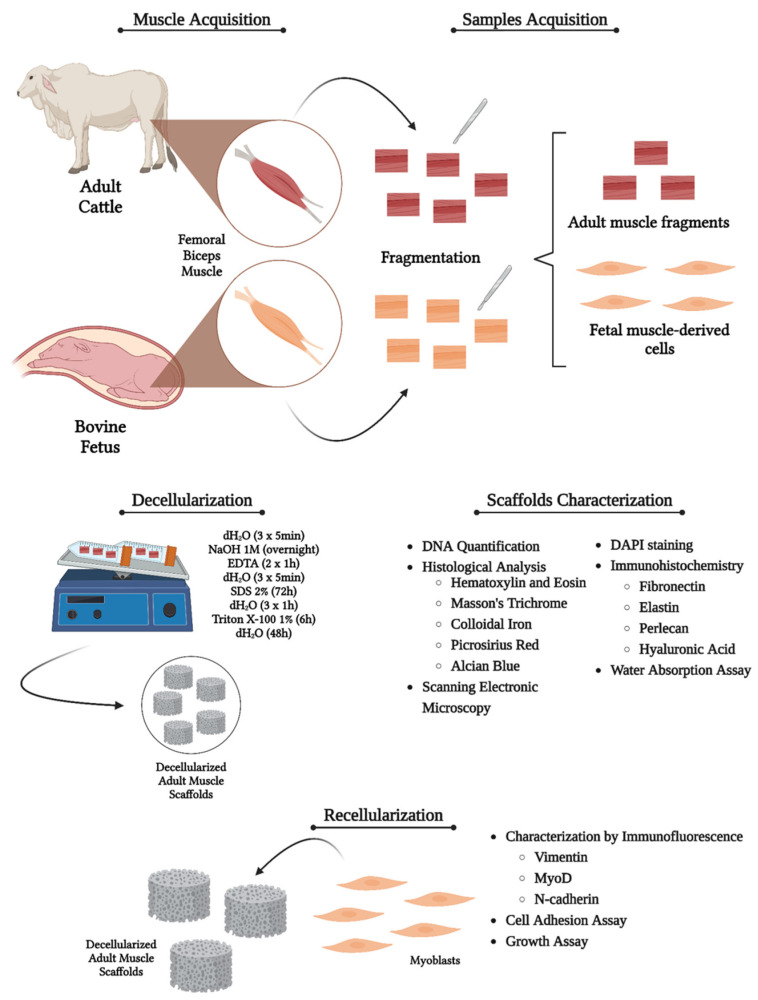
Experimental design of the study representing the steps from sample acquisition to recellularization analysis. Created with BioRender.com.

**Figure 2 cells-13-00688-f002:**
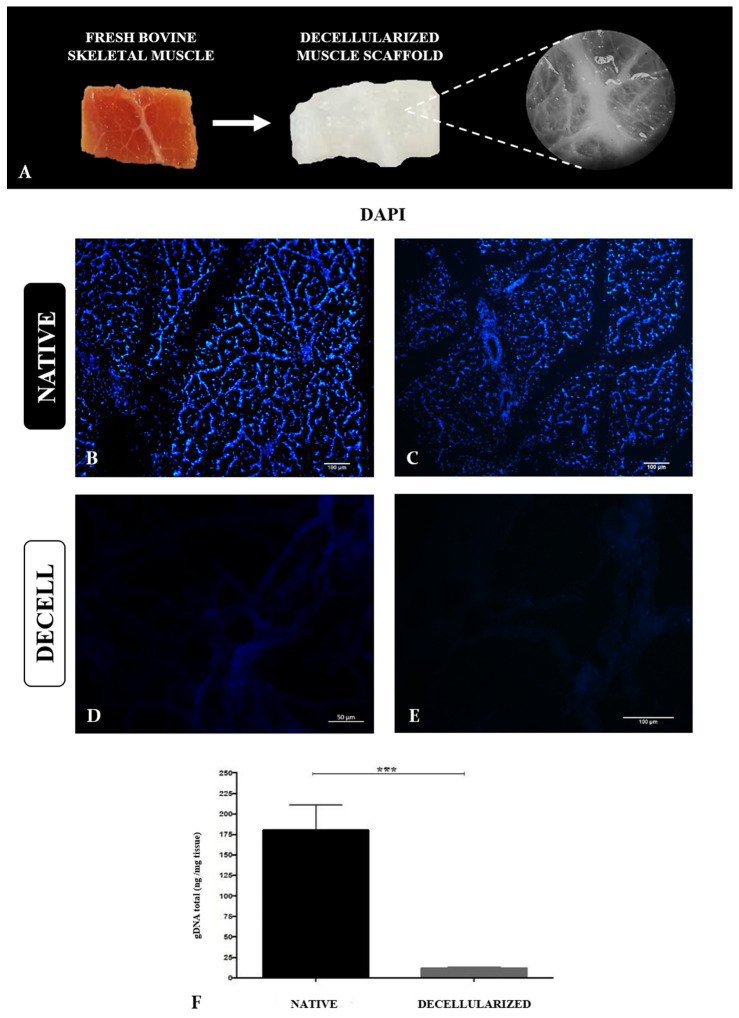
Decellularization of bovine skeletal muscle and production of muscle scaffolds. Photomicrography of bovine skeletal muscle before and after decellularization, highlighting the preservation of macroscopic tissue structures (**A**). DAPI staining of native (**B**,**C**) and decellularized samples (**D**,**E**). Total genomic DNA quantification of native and decellularized muscle samples (**F**). *** *p* < 0.05. Scale bars: 100 µm (**B**–**E**).

**Figure 3 cells-13-00688-f003:**
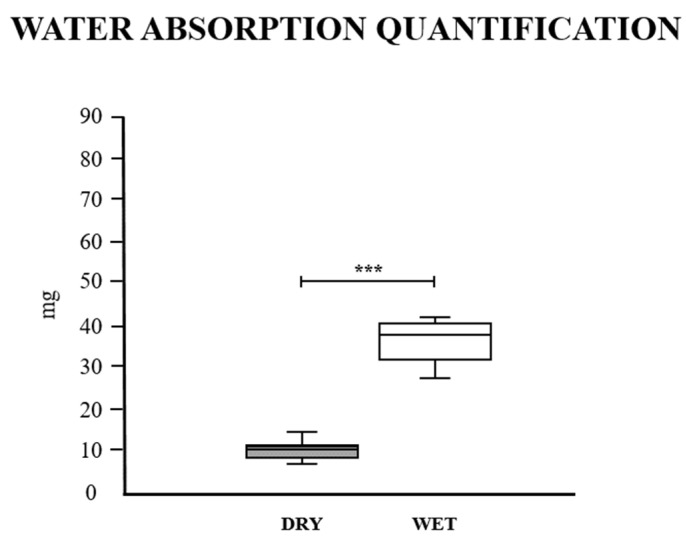
Water absorption assay to attest to the scaffolds’ fluid uptake ability and water retention capacity. Water absorption quantification, highlighting the weight difference in mg between dry and wet samples after 24 h. *** *p* < 0.05.

**Figure 4 cells-13-00688-f004:**
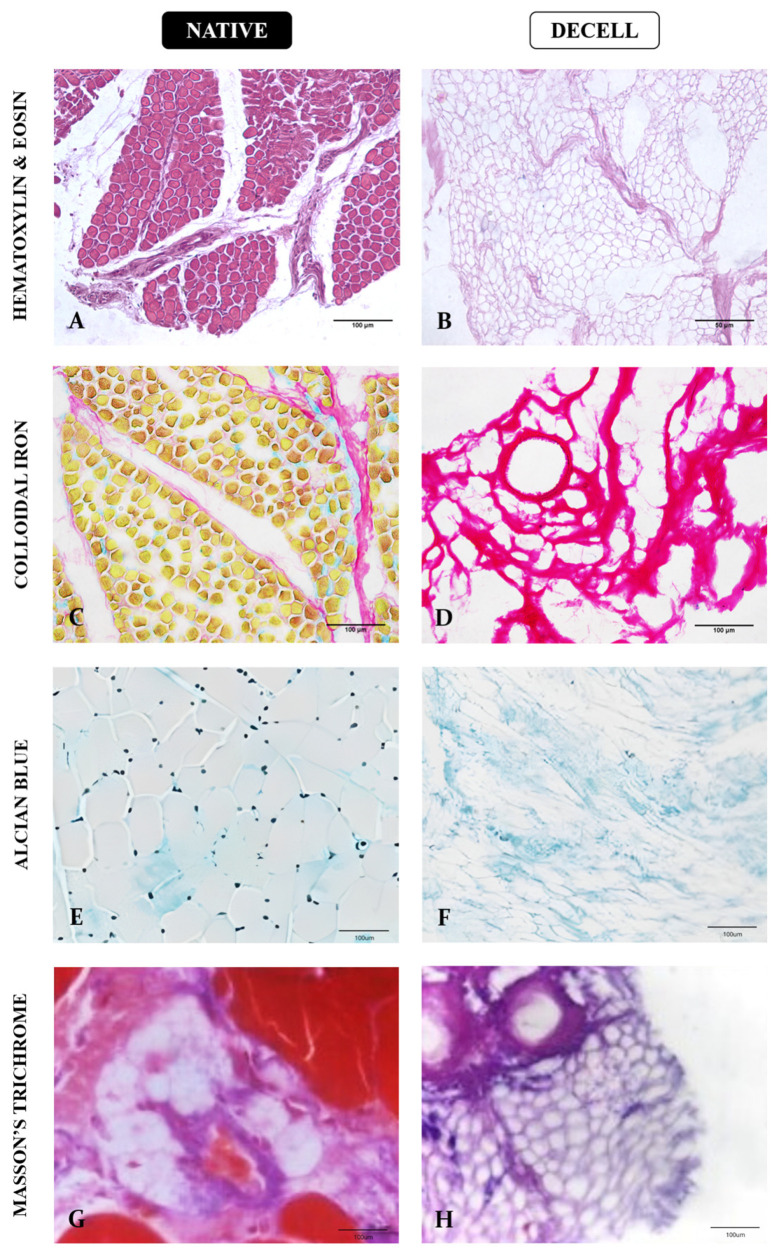
Histological analysis of extracellular matrix general components of native and decellularized muscle samples. Hematoxylin and Eosin staining to assess tissue general structure (**A**,**B**); Colloidal Iron staining to evaluate the proteoglycan content in pinkish red (**C**,**D**); Alcian blue staining to highlight general glycosaminoglycan content in light blue (**E**,**F**); Masson’s Trichrome staining for general collagen content in purple (**G**,**H**). Scale bars: 100 µm.

**Figure 5 cells-13-00688-f005:**
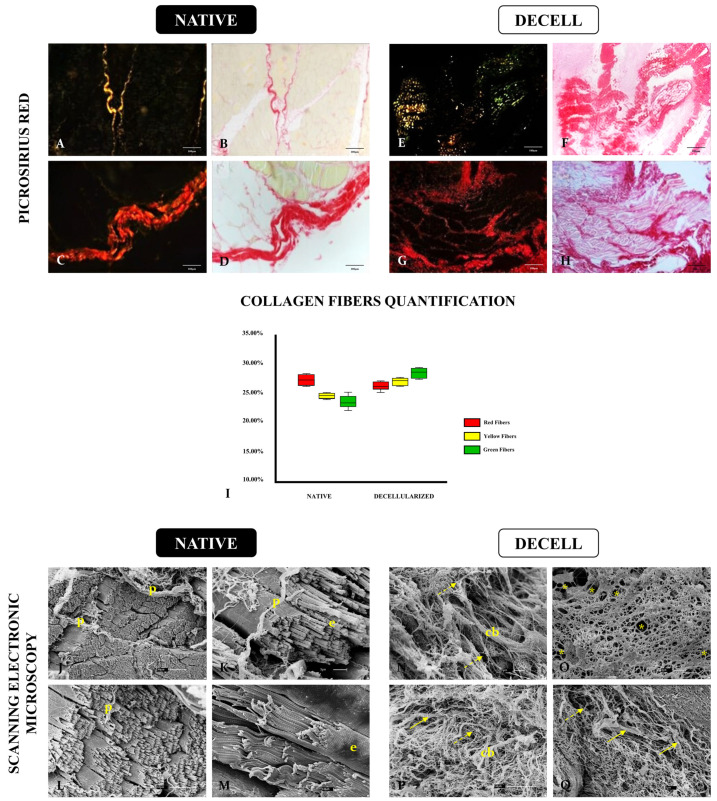
Histochemical and ultrastructural evaluation of native and decellularized bovine muscle samples. Picrosirius Red staining to highlight the different types of collagen fibers (**A**–**H**). Morphometric quantification of different collagen fibers according to their birefringence between native and decellularized samples (**I**). SEM micrographics to assess the 3D structure of native and decellularized muscle tissues (**J**–**Q**). Perimysium (p), endomysium (e), collagen bundle (cb). Full yellow arrows indicate thick fibers and dotted yellow arrows indicate thin fibers. Yellow asterisks represent the empty spaces in the tissue where myotubes were present. Scale bars: 100 µm (**A**–**H**), 30 µm (**J**), 10 µm (**L**,**M**,**P**) 3 µm (**K**,**N**,**O**,**Q**).

**Figure 6 cells-13-00688-f006:**
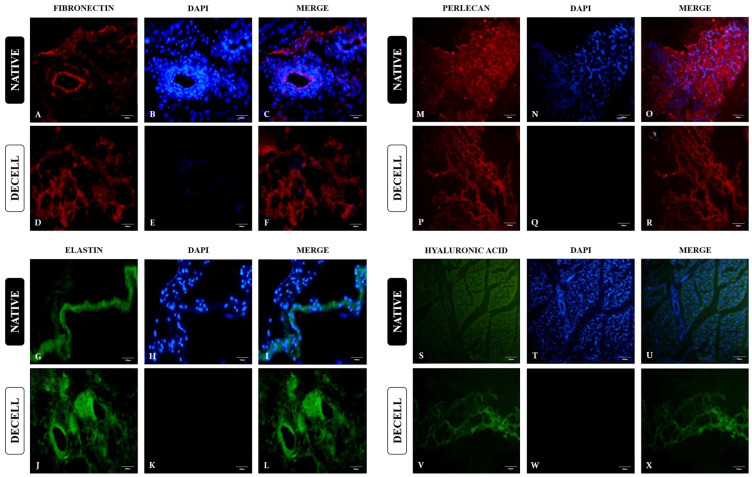
Immunolocalization of specific ECM proteins of bovine native and decellularized muscle samples. Immunofluorescence analysis of fibronectin (**A**–**F**), elastin (**G**–**L**), perlecan (**M**–**R**), and hyaluronic acid (**S**–**X**). Scale bars: 100 µm.

**Figure 7 cells-13-00688-f007:**
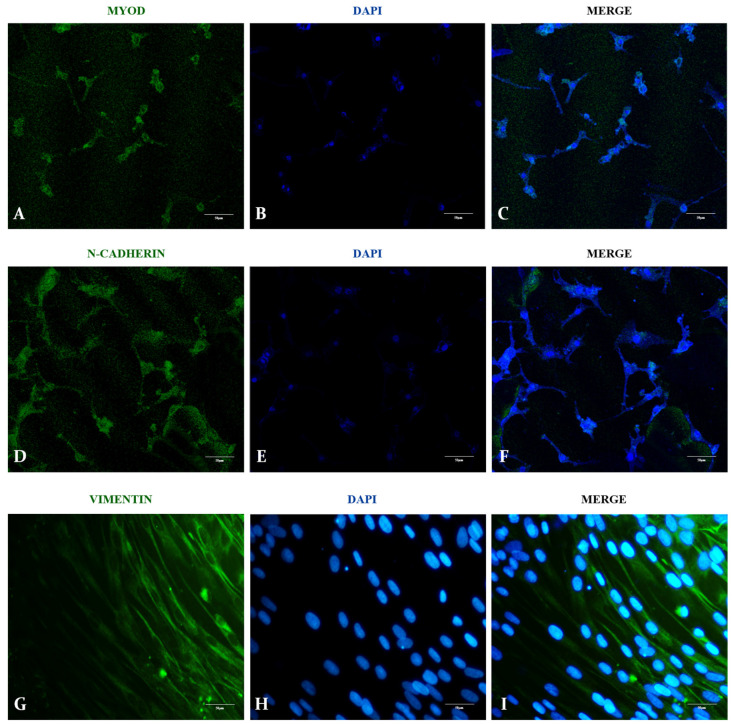
Characterization of fetal bovine myoblasts. The myoblast population was evaluated by immunocytochemistry for MyoD protein (**A**–**C**), N-cadherin (**D**–**F**), and vimentin (**G**–**I**). DAPI was used for nuclei visualization. Scale bars: 50 µm.

**Figure 8 cells-13-00688-f008:**
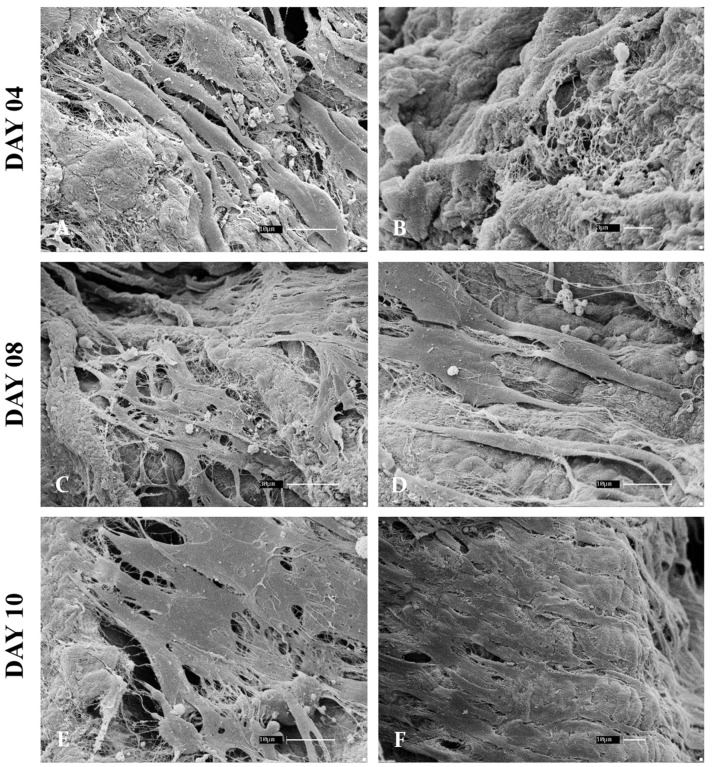
Scanning electronic micrographs to assess cellular adhesion performed with bovine myoblasts cultured on 3D bovine muscle scaffolds for 4, 8, and 10 days. On day 4, the cells were already anchored in the scaffolds (**A**), but there were still signs of uncovered matrix (**B**). On day 8, a more dense and interconnected cellular net is observed (**C**) and the anchored cells are beginning to form a layer over the scaffolds (**D**). On day 10, a more pronounced cell layer can be observed, which fills the majority of the surface of the scaffolds (**E**,**F**). Scale bars: 30 µm (**C**), 10 µm (**A**,**D**,**E**,**F**) 3 µm (**B**).

**Table 1 cells-13-00688-t001:** Antibodies used in the study.

Antibody	Dilution	Catalog Number
Elastin	1:100	ab21607
Fibronectin	1:100	ab6328
Hyaluronic Acid	1:50	PA1-85561
Perlecan	1:20	ab2501
Vimentin	1:1000	ab137321
N-Cadherin	1:1000	ab207608
MyoD	1:1000	ab16148

## Data Availability

The data presented in this study are available in this article.

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
