# Peer review of "Decellularized Bovine Skeletal Muscle Scaffolds: Structural Characterization and Preliminary Cytocompatibility Evaluation"

_cells, 2024, doi:10.3390/cells13080688_

Round 1

Reviewer 1 Report

Comments and Suggestions for Authors

Please find my comments and suggestions for authors in the attached pdf file.

Comments on the Quality of English Language

The English language is generally understandable, but extensive language editing is needed.

Author Response

April 2nd, 2024.

Editor-in-Chief

Cells

Dear editor,

Please find enclosed the revised version of our manuscript entitled “Decellularized Bovine Skeletal Muscle Scaffolds: Structural Characterization and Preliminary Cytocompatibility Evaluation” submitted to Cells.

We addressed all of the reviewer 1’s comments, which were extremely relevant for improving our manuscript. A point-by-point answer is below addressed:

Reviewer #1: ‘The abstract is nicely structured, but in my opinion it would benefit from expressing that what was novel in this study compared to previous literature.’

Answer: Thank you for the suggestion. We added this information in the abstract.

Reviewer #1: ‘The materials and methods seem to be somewhat comprehensively explained. However, the numbers of parallels seem to be missing from this section. Additionally, some other major omissions are described below.’

Answer: Thank you for the suggestion. We made changes according to your suggestions.

Reviewer #1: ‘The results are nicely laid out. On several occasions, such as on lines 215-220, 232-236, and several other locations, the text would in my opinion be more appropriately placed in the materials and methods section than in results.’

Answer: Thank you for the suggestions. We added in the Results section to correlate methodology with our experimental findings, which in our conception bring back the methodological design without being repetitive.

Reviewer #1: ‘As far as I understand, this is the first time that bovine muscles have been produced as scaffolds for human muscle tissue regeneration. I think that the discussion would generally benefit from assessing how these bovine muscle scaffolds compare to similar scaffolds or porcine, or even human, origin. E.g., have researchers been able to preserve the myotubes in those other decellularized scaffolds? If there are differences to the results obtained in this article, would it be because of the different origin of the tissue, or was there possibly something in the preparation methods that could have caused the differences?’

Answer: Thank you for the suggestion. We added more information about this topic in the Discussion section for your appreciation.

Reviewer #1: ‘The conclusions are nice and short. However, I am not convinced that “myoblast cell adhesion and growth after 10 days of culture” on the scaffolds was sufficiently proven in the results data.’

Answer: Thank you for the suggestion. We altered this in the Conclusions for ‘allowing myoblast cell adhesion and interaction with the scaffolds after 10 days of culture.’, which is more suitable and expressed by the results.

Reviewer #1: ‘Regarding the cell studies, I do not understand why the authors have cultured cells on the scaffolds and only characterized the scaffolds with SEM and not any other analyses. To me, this seems like the gravest omission in this paper. I would perhaps rather leave out the cellular assays entirely, as they do not bring much credible information about the scaffolds, or either develop them further with more tests. As this is a preliminary report, also some other relevant data is missing, such as mechanical characterization, but that will (probably) be reported in a later paper, and that is fine for me.’

Answer: Thank you for the comment. We agree completely that cellular assays are not complete, that is why we focused on the scaffolds characterization and put some preliminary cellular assays. The main reason was budgetary and we already published papers with biomaterials that only SEM analysis proved the interaction between the scaffold and cells (Santos Silva T, Melo Soares M, Oliveira Carreira AC, et al. Biological Characterization of Polymeric Matrix and Graphene Oxide Biocomposites Filaments for Biomedical Implant Applications: A Preliminary Report. Polymers (Basel). 2021;13(19):3382. Published 2021 Sep 30. doi:10.3390/polym13193382) and (Almeida GHDR, da Silva-Júnior LN, Gibin MS, et al. Perfusion and Ultrasonication Produce a Decellularized Porcine Whole-Ovary Scaffold with a Preserved Microarchitecture. Cells. 2023;12(14):1864. Published 2023 Jul 15. doi:10.3390/cells12141864). SEM analysis may not show functional data, but it does show if the cells were able to adhere and establish in the scaffolds, which infer cytocompatibility. Apoptosis and cell death morphology are very characteristic and evident by SEM analysis. Our data demonstrate that cells are anchored in the scaffolds and after 10 days, there is a visible difference in cell difference. More experiments are required to attest this, but as preliminary data, our findings indicate that.

Reviewer #1: ‘- In the two first lines in the introduction (lines 36-37), it is mentioned twice that skeletal muscles provide mobility. Even this reviewer remembers what was said on the previous line.’

Answer: We apologized for the mistake. We corrected that.

Reviewer #1: ‘-On line 51, the authors state that autologous muscle grafts may lead to the death of the patient. I believe that this has not been reported and that it is not stated in the original reference (ref. 7).’

Answer: Thank you for the suggestion. We changed the sentence.

Reviewer #1: ‘- In fact, in ref 7, it is stated there that the problems with autografts relate to “donor site scarcity, morbidity, and pain”. These problems are very different from what the authors write (“graft failure, prolonged surgical exposure time, development of opportunistic infections and necrosis of the grafted tissue and death”).’

Answer: Thank you for the suggestion. We added another reference to complement the sentence.

Reviewer #1: ‘-The sentence on lines 103-106 lacks a verb.’

Answer: Thank you for the suggestion. We changed the sentence.

Reviewer #1: ‘-The sentence on line 122 lacks a method or action.’

Answer: Thank you for the suggestion. We changed the sentence.

Reviewer #1: ‘-The 24-hour immersion is explained twice on lines 174 and 177.’

Answer: Thank you for the suggestion. We changed the sentence.

Reviewer #1: ‘Line 99 - What was the reason for using fetal myoblasts in the cell tests? The authors stress the clinical translation potential in the paper, but in clinical scenarios, fetal cells would not be used.’

Answer: Thank you for the question. Myoblasts are progenitor cells that can be differentiated into mature muscle fibers. There several studies that already used progenitor cells, embryonic or not, as satellite cells, mesenchymal stem cells or even iPSC, which recapitulate embryonic and fetal phenotypes, to perform muscle regeneration. These cells are highly sensitive and susceptible to apoptosis in non-suitable conditions and substrates. These myoblasts were available on our lab and based on progenitor cells therapy and their potential to be associated to muscle scaffolds to potentially generate biomimetic constructs, we chose to use them as a preliminary assay. Our next step is to associate these scaffolds with stem cells and other progenitor cells and evaluate their interaction with the scaffolds.

Reviewer #1: ‘Line 103 - This part would need a lot of clarification: How were these samples obtained? From which part of the muscles were they obtained? What was their shape? What size were these samples? How many samples were obtained totally? How many samples per fetus or adult? What was the age of the fetuses and the adult cows? In the results section, it is said’

Answer: Thank you for the suggestion. We added the missing information in the Materials and Methods section and, as mentioned above, part of this information is in the Results section.

Reviewer #1: ‘Lines 121 and 201- This is probably an error caused by the automated formatting program, but it would be good to make sure that the section titles are not blended with the section numbering in the final version.’

Answer: Thank you for the suggestion.

Reviewer #1:Line 171 - What was the shape of the samples?’

Answer: We added the information in the text. The samples were square-shape.

Reviewer #1: ‘Line 232 - Water absorption testing is good to perform, but I am not totally convinced that the water absorption is a good way of measuring porosity of the samples. The samples may swell during immersion, and it may also be possible that water is not penetrating into all pores. If the authors mean that they wanted to show that the scaffolds are porous, I am again not totally convinced that this is the best method, as water may possibly also penetrate through the pore walls.’

Answer: Thank you for the comment. We revised this concept, we agree with your concern, and based on another study evaluating decellularized scaffolds, we implied that the generated bovine muscle scaffolds have a high fluid uptake ability. We altered in the manuscript as well. (Bhoopathy J, Dharmalingam S, Sathyaraj WV, et al. Sericin/Human Placenta-Derived Extracellular Matrix Scaffolds for Cutaneous Wound Treatment-Preparation, Characterization, In Vitro and In Vivo Analyses. Pharmaceutics. 2023;15(2):362. Published 2023 Jan 20. doi:10.3390/pharmaceutics15020362).

Reviewer #1: ‘Line 234 - What does “protected from heat” mean? I think that it would be clearer if the authors could state the actual temperature.’

Answer: We added the information in the text. It was at 4°C in the fridge.

Reviewer #1: ‘Line 234 - What were the “monthly use tests”?’

Answer: Thank you for the suggestion. The text was altered. We meant that the samples could be used every month until six month after the drying, highlighting its durability.

Reviewer #1: ‘Line 237 - It would be beneficial if the authors could state the percentage of water absorption in writing.’

Answer: Thank you for the suggestion. We added the information in the text.

Reviewer #1: ‘Line 266 - The authors state “the absence of cells”. However, could those darker dots in fig. 4F be nuclei? According to DNA analysis, there is still some 5% cellular content left, and I would believe that they should be visible in histological analysis as well.’

Answer: Thank you for the comment. I would like to say that DNA analysis does not measure cellular content, but genomic DNA content, which are quite different, When the cells are lysed, DNA content may remain in the scaffolds, but the membrane and other cellular compartments not. What you may have seen in the image is a little drop of precipitated Alcian Blue stain, which happens during the histological process. Sometimes, an artifact appears, but should not be used to invalidate other solid data. The data that really shows the absence or presence of cells is DAPI and HE staining, which are clear of cells. We used Hematoxylin to counterstain Alcian Blue, but the real purpose of the staining is to highlight the GAG content, not nuclei.

Reviewer #1: ‘Line 281 - I do not understand why the authors refer to connective tissue and not muscle tissue.’

Answer: We are referring to the connective tissue that surround the muscle fibers, perimysium and endomysium. We altered in the text.

Reviewer #1: ‘Line 287 - The authors state that “both types of fibers remained preserved after the decellularization process”. Looking at figure 5 A-H and comparing the “native” and “decell” sides, to me the native fibers seem clearly more organized than the decell fibers, with more distinct features. Could the authors comment this in the manuscript?’

Answer: Thank you for the suggestion. We added a possible explanation for this finding.

Reviewer #1: ‘Lines 325 and onwards - Why were no quantitative measurements made already here for the cell cultures?’

Answer: Thank you for the question. The cellular data are related to a pilot assays that was performed to attest the ability of the scaffolds to allow cell adhesion and establishment after a determined period. This data was added as a preliminary result, showing the potential of bovine muscle scaffolds to support cell culture. Further studies will deep investigate the influence of these scaffolds on cell behavior and differentiation. We are waiting for a grant result to perform further quantitative assays and these results are preliminary data for the proposal submission.

Reviewer #1: ‘Line 349 - Maybe it is just me, but I struggle to see an increase in cell numbers between days 4 and 10 in the SEM images, although the authors state that “it is notable an increase on cell density on the scaffolds”.’

Answer: Thank you for the comment. We added more information to support this finding, which is preliminary and requires more analysis.

Reviewer #1: ‘Line 381 - The authors state that the protocol was effective to remove cellular content. However, according to the total genomic DNA quantification results, there was still approximately 5% DNA left in the decellularized matrices. In other papers, much higher reductions have been reported. I would like to read more in-depth discussion on this topic.’

Answer: According to Crapo et al. 2011 and reviewed by Gilpin et al. 2017, pioneers in decellularization, the criteria for an effective decellularized protocol is based on: ‘the decellularized ECM must have (1) less than 50 ng double-stranded DNA (dsDNA) per mg ECM dry weight, (2) less than 200 bp DNA fragment length, and (3) no visible nuclear material by 4′,6-diamidino-2-phenylindole (DAPI) staining.’ We performed DNA quantification that revealed less than 25 50 ng double-stranded DNA (dsDNA) per mg ECM dry weight and absence of nuclei in DAPI staining. The amount of DNA varies according to each tissue type, but the criteria is valid for all of them. Therefore, based on that, our protocol filled the decellularized efficiency criteria.

References used:

- Gilpin A, Yang Y. Decellularization Strategies for Regenerative Medicine: From Processing Techniques to Applications. Biomed Res Int. 2017;2017:9831534. doi: 10.1155/2017/9831534. Epub 2017 Apr 30. PMID: 28540307; PMCID: PMC5429943.

- Crapo P. M., Gilbert T. W., Badylak S. F. An overview of tissue and whole organ decellularization processes. Biomaterials. 2011;32(12):3233–3243. doi: 10.1016/j.biomaterials.2011.01.057.

Reviewer #1: ‘Line 382 - As far as I understand, the authors state here that their protocol preserved the muscle tissue structure. However, from the SEM figures (5 J-Q) it seems that the structure was not that well preserved. On lines 299-300, even the authors themselves write that “after the decellularization, the native muscle structure becomes disorganized”. There seems to be a conflict between the results and the discussion.’

Answer: Thank you for the comment. We would like to clarify that disorganization and degradation are completely different concepts. In the absence of cells, every connective tissue may lose their original rigid structure, once the cell-ECM adhesions are gone and the strength played by cell anchoring is no longer there. Said that, the main biological principle for a decellularized scaffold is to have its composition preserved and no signs of degradation, that is, destroyed collagen and elastin fibers, significant loss of GAGs and glycoproteins. Our results demonstrated that after the decellularization, the main ECM components remained, which implies that they are able to perform molecular signaling in the presence of reseeded cells. The skeletal muscle tissue has a higher proportion of cells compared to ECM, differently of other tissues such as cartilage, bone, uterus and ovary, which means that in the absence of cells and their shear stress forces, the fibers will suffer a loosening, which is reflected in collagen self-assembly. Based on that, our information is accurate and does not imply that ECM remained exactly as in the native tissue, but its composition and 3D structure remained preserved, which can favor cell adhesion and repopulation. The regenerative ability of the scaffold is to allow functionality, not recapitulate the exactly morphology of native tissue, but allow tissue remodeling to restore function.

I hope you find the revised version of our manuscript suitable for publication. Thank you in advance for your consideration.

Sincerely,

Prof. Dr. Rose Eli Grassi Rici

Faculty of Veterinary Medicine and Animal Science

University of São Paulo

Reviewer 2 Report

Comments and Suggestions for Authors

The work entitled “Decellularized Bovine Skeletal Muscle Scaffolds: Structural Characterization and Preliminary Cytocompatibility Evaluation”  provides a lot of information on the decellularization matrix properties compared to native tissue.  Overall, the work is scientifically sound. However, I have some suggestions before the manuscript has considered to be publish.

Skeletal musculature matrix is an highly orientation specific tissue,  however, according to the results, the matrix loses its anisotropic characterisation. Thus how to maintain the tissue specific biomechnical properties. The authors have to provides a clear overview of the problem statement regarding preparation and application of decellularized Scaffolds and highlight the importance of the need for using Bovine Skeletal Muscle tissue, which is crucial to explore the current work. The novelty aspect should be explained in details, as lot of literature is already available for the Characterization and evaluation of decellularized Scaffolds.

In general, the main components of connective tissue ECM are collagen fibres and GAG. I think as many decellularization methods are established, it is a basic requirement that the decellularized matrix has the same composition as the native tissue. But the protein and GAG content varies from tissue to tissue. So do the authors compare the quantity of main components of native and decellularized matrix?

Figure 2, D-F, results show incomplete decellularization. Therefore, my question is how to control the quality of each batch of decellularized scaffolds produced?

Figure 3, are the samples A and B the same? Or is it different? The dry samples shown in A appear larger in size than the wet samples shown in B.

Line 284, please give some explanations or cite related references about  "Under polarized light microscopy, the distinction of the different collagen types by fiber diameter is possible, once fibers with reddish and yellowish tons are related to thick collagen fibers and the greenish ones are related to thin fibers.”

Line 301, I don't understand what "the presence of coarse and fine fibers and the maintenance of a rounded structure" means. Please indicate which fibers are thick and which are thin in the images.

Figure 8, Please add explanation of subfigures in Figure 8A to 8F.

The scale bar in all figures is not clear enough. 

Author Response

April 2nd, 2024.

Editor-in-Chief

Cells

Dear editor,

Please find enclosed the revised version of our manuscript entitled “Decellularized Bovine Skeletal Muscle Scaffolds: Structural Characterization and Preliminary Cytocompatibility Evaluation” submitted to Cells.

We addressed all of the reviewer 2’s comments, which were extremely relevant for improving our manuscript. A point-by-point answer is below addressed:

Reviewer #2: ‘Skeletal musculature matrix is a highly orientation specific tissue, however, according to the results, the matrix loses its anisotropic characterization. Thus how to maintain the tissue specific biomechnical properties. The authors have to provide a clear overview of the problem statement regarding preparation and application of decellularized Scaffolds and highlight the importance of the need for using Bovine Skeletal Muscle tissue, which is crucial to explore the current work. The novelty aspect should be explained in details, as lot of literature is already available for the Characterization and evaluation of decellularized Scaffolds.’

Answer: Thank you for the suggestion. We added more information about this topic in the Introduction and Discussion sections for you appreciation and evaluation.

Reviewer #2: ‘In general, the main components of connective tissue ECM are collagen fibers and GAG. I think as many decellularization methods are established, it is a basic requirement that the decellularized matrix has the same composition as the native tissue. But the protein and GAG content varies from tissue to tissue. So do the authors compare the quantity of main components of native and decellularized matrix?’

Answer: We performed a qualitative evaluation between native and decellularized samples and we attested that there was a similar distribution pattern in both groups, inferring the main ECM components preservation.

Reviewer #2: ‘Figure 2, D-F, results show incomplete decellularization. Therefore, my question is how to control the quality of each batch of decellularized scaffolds produced?’

Answer: Thank you for the comment, but such figures do not show incomplete decellularization. For colloidal iron staining, muscle fibers are stained in yellow and proteoglycan content in pinkish red. In Figure 2D, it is noteworthy that there is no yellow staining in it. In Figure 2F, these intense blue microdots are not nuclei. They can be related to a high concentrated point of GAGs or precipitated Alcian blue staining, which may occur as a histological artifact. These nuances does not disqualify the other data that attested the decellularization efficiency. Regarding batch control quality, an assay that is mandatory for every decellularization is DNA quantification and DAPI staining that will corroborate the decellularization. With only these two data, the quality control of the decellularization process is done according to Badylak’s research group, pioneer in decellularization (Crapo P. M., Gilbert T. W., Badylak S. F. An overview of tissue and whole organ decellularization processes. Biomaterials. 2011;32(12):3233–3243. doi: 10.1016/j.biomaterials.2011.01.057). For every decellularized samples, at least both assays were carried out.     

Reviewer #2: ‘Figure 3, are the samples A and B the same? Or is it different? The dry samples shown in A appear larger in size than the wet samples shown in B.’

Answer: Thank you for the comment. We altered the figure to avoid any subjective questions, once we measured the samples and generated the graphic based on quantitative data. The subject aspect of the samples may overshadow the results.

Reviewer #2: ‘Line 284, please give some explanations or cite related references about  "Under polarized light microscopy, the distinction of the different collagen types by fiber diameter is possible, once fibers with reddish and yellowish tons are related to thick collagen fibers and the greenish ones are related to thin fibers.”’

Answer: Thank you for the suggestion. We added three references to the manuscript related to that, including the original one published by Junqueira et al., 1978.

Reviewer #2: ‘Line 301, I don't understand what "the presence of coarse and fine fibers and the maintenance of a rounded structure" means. Please indicate which fibers are thick and which are thin in the images.

Answer: Thank you for the comment. We replaced the expression ‘round structures’ for empty circular spaces’ to indicate where the myotubes were located. We pointed in the figure the thick and thin fibers with different arrows and explained in the caption.

Reviewer #2: ‘Figure 8, Please add explanation of subfigures in Figure 8A to 8F.’

Answer: Thank you for the suggestion. We added explanation to the images.

Reviewer #2: ‘The scale bar in all figures is not clear enough.’

Answer: Thank you for the comment. We adjusted the scale bars and we added the information in the end of the figures captions.

I hope you find the revised version of our manuscript suitable for publication. Thank you in advance for your consideration.

Sincerely,

Prof. Dr. Rose Eli Grassi Rici

Faculty of Veterinary Medicine and Animal Science

University of São Paulo

Round 2

Reviewer 1 Report

Comments and Suggestions for Authors

I would like to thank the Authors for all the clarifications that you have contributed with.

A quick check reveals that on line 105, it is now stated that the samples were two-dimensional (2 cm x 2 cm), although I believe that the samples had a measurable thickness as well.

On line 180, I suppose the sample shape was cubic? A square shape is, as far as I understand, two-dimensional.

On line 469, I do not comprehend the meaning of "(...)".

If these are addressed (along with improvement of the English language), I would be happy to recommend publication of this paper.

Comments on the Quality of English Language

Despite some improvements, there are still several deficiencies in the English language, such as a missing 'are' on line 44. I also believe that many sentences could be written in a clearer manner.

I still believe that the language should be improved.

Author Response

April 8th, 2024.

Editor-in-Chief

Cells

Dear editor,

Please find enclosed the revised version of our manuscript entitled “Decellularized Bovine Skeletal Muscle Scaffolds: Structural Characterization and Preliminary Cytocompatibility Evaluation” submitted to Cells.

We addressed the entire second round of reviewer 1’s comments, which were extremely relevant for improving our manuscript. A point-by-point answer is below addressed:

Reviewer #1: ‘A quick check reveals that on line 105, it is now stated that the samples were two-dimensional (2 cm x 2 cm), although I believe that the samples had a measurable thickness as well.’

Answer: Thank you for the suggestion. It was a typing mistake, we corrected it.

Reviewer #1: ‘On line 180, I suppose the sample shape was cubic? A square shape is, as far as I understand, two-dimensional.

Answer: Thank you for the suggestion. It was a typing mistake, we corrected it as well.

Reviewer #1: ‘On line 469, I do not comprehend the meaning of "(...)".’

Answer: We apologized for the mistake, we added the missing references.

Reviewer #1: ‘Despite some improvements, there are still several deficiencies in the English language, such as a missing 'are' on line 44. I also believe that many sentences could be written in a clearer manner. I still believe that the language should be improved.’

Answer: We revised the entire manuscript along with a Native speaker expert and performed alterations. Thank you for all the suggestions.

I hope you find the revised version of our manuscript suitable for publication. Thank you in advance for your consideration.

Sincerely,

Prof. Dr. Rose Eli Grassi Rici

Faculty of Veterinary Medicine and Animal Science

University of São Paulo

Reviewer 2 Report

Comments and Suggestions for Authors

The authors don’t provide an explantation for the question related to “Skeletal musculature matrix is an highly orientation specific tissue,  however, according to the results, the matrix loses its anisotropic characterisation. Thus how to maintain the tissue specific biomechnical properties”. Furthermore, decellularization is a process by which any natural tissue can be converted into a material, so if porcine-derived biomaterials prove successful as materials, the bioethical shortcomings of religious beliefs cannot be a key reason in performing similar procedures on tissues from different animal species. Are there other advantages that only bovine skeletal muscle can offer from a biomaterials perspective?  My main question is, since decellularization is a well-established technology and porcine-sourced biomaterials have also been proven successful, what are the crucial reasons for evaluating bovine-sourced biomaterials, considering they are both large mammals with similar anatomical structures?

The authors don’t address my question. Lots of qualitative evaluation between native and decellularized samples to demonstrate similar distribution pattern in both groups indicating the main ECM components preservation. My question is do the authors compare the contents of main components of native and decellularized matrix?

Figure 2 is an image of  DAPI staining, so I cannot understand this sentence “For colloidal iron staining, muscle fibers are stained in yellow and proteoglycan content in pinkish red. In Figure 2D, it is noteworthy that there is no yellow staining in it. In Figure 2F, these intense blue microdots are not nuclei. “

Figure 3, missing A.

Figure 5, (1) please point “empty circular spaces’ in the images. (2) In image O, no any arrows shows. (3) In caption,  1 um (?), missing information.

Figure 8,  please add explanation of subfigures in Figure 8A to 8F such as the magnification and what are the differences between A and B, C and D, E and F?

Line 469, (….)  missing information. 

The scale bar in all figures is still not clear enough.

Author Response

April 8th, 2024.

Editor-in-Chief

Cells

Dear editor,

Please find enclosed the revised version of our manuscript entitled “Decellularized Bovine Skeletal Muscle Scaffolds: Structural Characterization and Preliminary Cytocompatibility Evaluation” submitted to Cells.

We addressed the entire second round of the reviewer 2’s comments, which were extremely relevant for improving our manuscript. A point-by-point answer is below addressed:

Reviewer #2: ‘The authors do not provide an explanation for the question related to “Skeletal musculature matrix is a highly orientation specific tissue, however, according to the results, the matrix loses its anisotropic characterization. Thus how to maintain the tissue specific biomechanical properties”.

Answer: It is important to highlight that, in the absence of cells, every connective tissue may lose their original rigid structure, once the cell-ECM adhesions are gone and the strength played by cell anchoring is no longer there. Said that, the main biological principle for a decellularized scaffold is to have its composition preserved and no signs of degradation, that is, destroyed collagen and elastin fibers, significant loss of GAGs and glycoproteins. Our results demonstrated that after the decellularization, the main ECM components remained, which implies that they are able to perform molecular signaling in the presence of reseeded cells. The skeletal muscle tissue has a higher proportion of cells compared to ECM, differently of other tissues such as cartilage, bone, uterus and ovary, which means that in the absence of cells and their shear stress forces, the fibers will suffer a loosening, which is reflected in collagen self-assembly. Based on that, our information is accurate and does not imply that ECM remained exactly as in the native tissue, but its composition and 3D structure remained preserved, which can favor cell adhesion and repopulation. The regenerative ability of the scaffold is to allow functionality, not recapitulate the exactly morphology of native tissue, but allow tissue remodeling to restore function. Once the cells are muscle reseeded on the scaffolds, due to molecular signs that are tissue-specific and are responsible for tissue memory may guide the muscle cells orientation. In addition, the application of the generated biomaterials is not restricted to a scaffold use, but it may be used as substrate to develop ECM-based hydrogels that can be applied in 3D bioprinted constructs or even as a bioactive components to be incorporated in other pharmaceutical forms to induce tissue repair in vivo.

Reviewer #2: ‘Furthermore, decellularization is a process by which any natural tissue can be converted into a material, so if porcine-derived biomaterials prove successful as materials, the bioethical shortcomings of religious beliefs cannot be a key reason in performing similar procedures on tissues from different animal species. Are there other advantages that only bovine skeletal muscle can offer from a biomaterials perspective?  My main question is, since decellularization is a well-established technology and porcine-sourced biomaterials have also been proven successful, what are the crucial reasons for evaluating bovine-sourced biomaterials, considering they are both large mammals with similar anatomical structures?’

Answer: It is important to understand that the success of porcine based-biomaterials does not discredit the research for novel biological sources of biomaterials as bovine, ovine and other livestock animals. There are no studies that compare muscle scaffolds from porcine and bovine skeletal muscle tissues. Nevertheless, there are data that reports allergic reactions from porcine collagen-derived biomaterials, which reveals the necessity of biomaterials from other animal sources in order to overcome these immunological drawbacks in specific cases that the porcine biomaterial is not an option. Another reason is the muscle source for the biomaterials production. So far, studies involving decellularized porcine muscle obtained their samples from limited muscles in terms of mass volume and fiber strength as psoas muscle, rectus abdominal muscle and tibialis posterior. These muscles are a limited source for muscle-based biomaterials. This motivated our group to search for a more suitable source from bovine tissue. For that reason, the samples acquisition were standardized from the femoral biceps muscle due to its representative size and weight in the animal, shear force being between the threshold of slightly soft and slightly hard, and its low to medium cost financially and significant water retention capacity. I hope this have elucidated better the reasons for this research.

Reviewer #2: ‘The authors don’t address my question. Lots of qualitative evaluation between native and decellularized samples to demonstrate similar distribution pattern in both groups indicating the main ECM components preservation. My question is do the authors compare the contents of main components of native and decellularized matrix?’

Answer: Unfortunately, we did not have any biochemical assay kits to compare the contents between the groups. In order to overcome this limitation, we performed a broad qualitative comparison between the groups to infer the ECM components preservation, evaluating through histology and immunohistochemistry of the components. A future step is to perform a proteomic analysis of the scaffolds for a deeper evaluation of the main ECM components.  

Reviewer #2: ‘Figure 2 is an image of DAPI staining, so I cannot understand this sentence “For colloidal iron staining, muscle fibers are stained in yellow and proteoglycan content in pinkish red. In Figure 2D, it is noteworthy that there is no yellow staining in it. In Figure 2F, these intense blue microdots are not nuclei. “’

Answer:  We apologize for the mistake. We were referring to Figure 4.

Reviewer #2: ‘Figure 3, missing A.’

Answer: Thank you for the suggestion. We corrected that.

Reviewer #2: Figure 5, (1) please point “empty circular spaces’ in the images. (2) In image O, no any arrows shows. (3) In caption, 1 um (?), missing information.’

Answer: Thank you for the suggestions. We added asterisks in Figure 5O to highlight the empty circular spaces, where the myotubes once were. Figure 5O is not the figure that we point the different calibers of fibers. They are better represented in Figures 5N, P and Q. We corrected the caption

Reviewer #2: ‘Figure 8,  please add explanation of subfigures in Figure 8A to 8F such as the magnification and what are the differences between A and B, C and D, E and F?’

Answer: Thank you for the suggestion. We add a better explanation in Figure 8 caption for a better understanding.

Reviewer #2: ‘Line 469, (….) missing information.’

Answer: We apologized for the mistake. We added the missing references.

Reviewer #2: ‘The scale bar in all figures is still not clear enough.’

Answer: We improved the quality of all figures as well as the scale bars. These are the original figures with no alterations. We hope it is suitable now.

I hope you find the revised version of our manuscript suitable for publication. Thank you in advance for your consideration.

Sincerely,

Prof. Dr. Rose Eli Grassi Rici

Faculty of Veterinary Medicine and Animal Science

University of São Paulo